# Spanish HCMV Seroprevalence in the 21st Century

**DOI:** 10.3390/v16010006

**Published:** 2023-12-19

**Authors:** Pablo Álvarez-Heredia, Irene Reina-Alfonso, José Joaquín Domínguez-del-Castillo, Fakhri Hassouneh, Carmen Gutiérrez-González, Alexander Batista-Duharte, Ana-Belén Pérez, Fernando Sarramea, María José Jaén-Moreno, Cristina Camacho-Rodríguez, Raquel Tarazona, Rafael Solana, Juan Molina, Alejandra Pera

**Affiliations:** 1Immunology and Allergy Group (GC01), Maimonides Biomedical Research Institute of Cordoba (IMIBIC)/University of Cordoba/Reina Sofia University Hospital, Av. Menendez Pidal s/n, 14004 Cordoba, Spain; irene.reina@imibic.org (I.R.-A.); fakhri.hassouneh@imibic.org (F.H.); carmen.gutierrez@imibic.org (C.G.-G.); alexander.batista@imibic.org (A.B.-D.); rsolana@uco.es (R.S.); juan.e.molina.alcaide@gmail.com (J.M.); 2Cardiovascular Pathology (GA09), Maimonides Biomedical Research Institute of Cordoba (IMIBIC)/University of Cordoba/Reina Sofia University Hospital, Av. Menendez Pidal s/n, 14004 Cordoba, Spain; jjdomdel@gmail.com; 3Microbiology Service, Reina Sofia University Hospital of Cordoba/Maimonides Biomedical Research Institute (IMIBIC)/CIBERINFEC, Av. Menendez Pidal s/n, 14004 Cordoba, Spain; anab.perez.jimenez.sspa@juntadeandalucia.es; 4Severe Mental Illness-Health Alerts (GA12), Maimonides Biomedical Research Institute of Cordoba (IMIBIC)/University of Cordoba/Reina Sofia University Hospital, Av. Menendez Pidal s/n, 14004 Cordoba, Spain; fscferro69@gmail.com (F.S.); mjjaen@uco.es (M.J.J.-M.); cristinacamachordz@gmail.com (C.C.-R.); 5Department of Morphological and Socio-Health Sciences, University of Cordoba, Av. Menendez Pidal s/n, 14004 Cordoba, Spain; 6Mental Health Clinical Management Unit, Reina Sofía University Hospital/ CIBERSAM, Av. Menendez Pidal s/n, 14004 Cordoba, Spain; 7Immunology Unit, Department of Physiology, University of Extremadura, 10003 Cáceres, Spain; rtarazon@unex.es; 8Department of Cell Biology, Physiology and Immunology, University of Cordoba, Av. Menendez Pidal s/n, 14004 Cordoba, Spain; 9Immunology and Allergy Service, Reina Sofia University Hospital of Cordoba, Av. Menendez Pidal s/n, 14004 Cordoba, Spain

**Keywords:** HCMV, seroprevalence, Spain, inflammation

## Abstract

Human cytomegalovirus (HCMV) is linked to age-related diseases like cardiovascular disease, neurodegenerative conditions, and cancer. It can also cause congenital defects and severe illness in immunocompromised individuals. Accurate HCMV seroprevalence assessment is essential for public health planning and identifying at-risk individuals. This is the first HCMV seroprevalence study conducted in the general Spanish adult population in 30 years. We studied HCMV seroprevalence and HCMV IgG antibody titres in healthy adult donors (HDs) and HCMV-related disease patients from 2010 to 2013 and 2020 to 2023, categorized by sex and age. We compared our data with 1993 and 1999 studies in Spain. The current HCMV seroprevalence among HDs in Spain is 73.48%. In women of childbearing age, HCMV seroprevalence has increased 1.4-fold in the last decade. HCMV-seropositive individuals comprise 89.83% of CVD patients, 69% of SMI patients, and 70.37% of COVID-19 patients. No differences in HCMV seroprevalence or HCMV IgG antibody titres were observed between patients and HDs. A significant reduction in Spanish HCMV seroprevalence among HDs was observed in 1993. However, women of childbearing age have shown an upturn in the last decade that may denote a health risk in newborns and a change in HCMV seroprevalence trends.

## 1. Introduction

Human cytomegalovirus (HCMV) is a herpesvirus that can infect individuals of any age and remain latent in the host for life, leading to periodic reactivation and potential transmission. HCMV can be spread through close contact with bodily fluids, such as saliva, urine, and breast milk, and can also be transmitted through blood transfusions, organ transplants, and sexual contact. HCMV infection is particularly dangerous for newborns and immunocompromised individuals, such as those with HIV/AIDS or organ transplant recipients. In immunocompetent individuals, HCMV infection typically remains asymptomatic, though rare cases of mononucleosis have been reported. Nevertheless, it is noteworthy that this infection can have a profound impact on the immune system [1,2,3]. HCMV is a key contributor to immunosenescence, which is the gradual deterioration of the immune system with age and includes changes in the phenotype and functions of both innate and adaptive cells. Moreover, this virus causes persistent low-level inflammation or “inflammaging”. This asymptomatic process will ultimately lead to the accumulation of senescent T cells and reduced immune function [4]. In the past decade, numerous researchers have studied this chronic inflammation and the accumulation of senescent cells driven by HCMV has been linked to age-related diseases such as cardiovascular disease, neurodegenerative diseases, autoimmune disease, and cancer [5,6,7]. Thus, HCMV chronic infection has gained importance due to its impact and contribution to the development of pathologies affecting the health of populations worldwide.

Recent studies have shown a global HCMV seroprevalence of 83% in the general population, with the highest prevalence in the Eastern Mediterranean region (90%) and the lowest in the European region (66%) [8]. These differences can be explained by the cultural divergence among European countries. In this sense, the social and cultural context of the Spanish population is characterized by a higher level of social activity compared to other European countries, and with older members of families cohabiting with young members. This is particularly prominent in the south of Spain, specifically in the region of Andalusia, which is the most populated region of Spain (17.8%). No previous studies have been conducted on HCMV seroprevalence in this region, and the latest studies in this regard are outdated [9,10]. Considering the region’s popularity as a tourist destination (30.7 million visitors in 2022 according to data from the regional Ministry of tourism, culture, and sport), and given the health consequences derived from HCMV infection, it is crucial to accurately determine the virus’s presence not only in Andalusia, but also in the rest of the Spanish territory. Our research group has conducted an analysis of the seroprevalence of this viral infection in healthy donors collected during two distinct periods of the 21st century: the first period was between 2010 and 2013 and the second period was between 2020 and 2023. The primary objective of this analysis was to provide an update on the healthy adult seroprevalence of HCMV in the region. Additionally, we aimed to examine the differences in seroprevalence between the two periods, which were a decade apart. Furthermore, we performed a comparative study investigating the presence of HCMV chronic infection and anti-HCMV IgG antibody titres among healthy controls and individuals with HCMV-associated inflammatory-based diseases, including cardiovascular diseases such as aortic stenosis, aortic or mitral pathology and coronary artery disease (CVD) [11,12], serious mental illness (SMI) [13,14], and the novel COVID-19 [15,16].

## 2. Materials and Methods

### 2.1. Study Design and Participants

Our study aims to investigate the variations in cytomegalovirus prevalence in the adult population and its correlation with various HCMV-related and inflammatory-based diseases. For this purpose, we collected blood samples from 252 healthy adult donors (>18 years) during two distinct periods, resulting in two cohorts: the first cohort was recruited between 2010 and 2013 (HD-1, *n* = 120), and the second cohort was recruited between 2020 and 2023 (HD-2, *n* = 132). Furthermore, to compare the seroprevalence of HCMV in our cohorts with the last available data regarding the general healthy Spanish population, performed in the 20th century (1993) [9], our analysis was restricted to individuals between the ages of 18 and 60 years. To track changes over time in HCMV seroprevalence among women of childbearing age (18–40 years), we performed a comparison with data from the previous Spanish studies conducted in the 20th century (1993 and 1999) [9,10]. Moreover, HD-2 was divided into three sub-cohorts: HD-A (CVD controls), HD-B (SMI controls), and HD-C (COVID-19 controls), where individuals were sex/age matched with each patient group (Figure 1). In addition to studying HCMV seroprevalence in the general adult population, our group also aimed to examine it in patients with specific medical conditions such as cardiovascular disease (CVD: aortic/mitral stenosis or coronary artery disease), serious mental illness (SMI: schizophrenia and bipolar disorder), and hospitalized and non-hospitalized COVID-19 patients. These patients’ data were collected between 2020 and 2023, reflecting the more recent period of our study. The eligibility criteria for all study participants included the absence of a history of cancer or autoimmune disease and no ongoing immunosuppressive treatment. Moreover, for healthy donors from both the 2010–2013- and 2020–2023-time frames, additional prerequisites included the absence of any cardiovascular or inflammatory-based diseases, serious mental illness, and previous infections with the SARS-CoV-2 virus. Individuals with cardiovascular disease were included if they had been specifically indicated for aortic valve replacement surgery. It was crucial that they exhibited no signs of mental illness or any other inflammatory-based disease and had no prior history of SARS-CoV-2 infection. Concerning individuals infected with SARS-CoV-2, they were categorized into two groups based on the severity of the infection, namely whether hospitalization was required or not. Lastly, individuals diagnosed with serious mental illness (SMI) were included if they met the diagnostic criteria for bipolar disorder according to the F31 ICD-10 classification, in any of its subtypes, or were diagnosed with schizophrenia following the F20 ICD-10 criteria. Detailed information regarding the inclusion and exclusion criteria for each cohort and sub-cohort are detailed in Table 1.

### 2.2. Sample Collection and HCMV Determination

Peripheral blood was collected from each participant using tubes containing heparin/lithium as an anticoagulant through venipuncture. Following collection, plasma samples were extracted from these tubes via centrifugation. These samples were then carefully preserved by freezing them at −80 °C until the day of analysis. For the HCMV determination, plasma samples were thawed and processed within 24 h. This analysis was carried out by the Microbiology Service of the Reina Sofía University Hospital using the chemiluminescent immunoassay technique (CLIA) to detect the presence of IgM and IgG antibodies specific to the cytomegalovirus (HCMV). This assay was conducted using the Liaison XL^®^ system, manufactured by Diasorin in Italy. Specifically, the Liaison HCMV IgM II and Liaison HCMV IgG II assays were utilized for IgM and IgG antibody detection, respectively.

### 2.3. Statistical Analysis and Data Processing

IBM Statistics SPSS 26 was utilized for the statistical analysis of the study. Descriptive statistics were employed to calculate the mean age, standard deviation, and percentage of HCMV-seropositive individuals. A weighted calculation of HCMV seroprevalence values was performed to compare our cohort of healthy donors with those from previous seroprevalence studies in Spain. Chi-square tests were conducted to compare recruitment periods, and between patients and controls. The Shapiro–Wilk test was used to determine a normal or non-normal distribution in HCMV IgG antibody titres. According to this, the Student’s *t*-test or Mann–Whitney U test was performed. Significant *p*-values were indicated. The GraphPad Prism software (version 8.0; GraphPad Software) was utilized for generating charts and plots of the study results.

## 3. Results

### 3.1. Study Population Characteristics

A total of 549 individuals were included in the study, comprising 252 healthy adult donors (≥18 years) and 297 patients with HCMV-associated inflammatory-based disease. The cohort was divided as follows: HD-1 (*n* = 120), HD-2 (*n* = 132), patients with cardiovascular disease (CVD) (*n* = 118), hospitalized COVID-19 patients (hCOVID-19, *n* = 78), non-hospitalized COVID-19 patients (nhCOVID-19 n = 30), and patients with serious mental illness (SMI) (*n* = 71), (Figure 1). The demographics of healthy donors and patients, including their age and sex, are presented in Table 2 and Table 3.

### 3.2. HCMV Latent Infection among Adult Spanish Population

Our study sample is comprised of two recruitment periods set in 2010–2013 (HD-1) and 2020–2023 (HD-2) (Figure 1). The HD-1 sub-cohort exhibited an HCMV seroprevalence of 70.83%, while the seroprevalence of the HD-2 sub-cohort was 73.48% (Table 2). These results indicate that there has been no significant change in the seroprevalence of HCMV over the last ten years. Additionally, the seroprevalence of HCMV within HD-1 and HD-2 was also calculated according to age (young, middle age, older). The results showed an ascending trend of HCMV seroprevalence with age in both periods. Upon conducting this analysis categorized by sex, no significant differences were observed between HD-1 and HD-2 cohorts independently of age (Table 2). 

Moreover, our data show a 1.4-fold increase in HCMV seroprevalence for women at childbearing age (18–40 years) from 2013 to 2023, contrary to the previous data reported by de Ory et al. [9,10]. It is noteworthy that the HCMV seroprevalence reported by de Ory et al. in 1993 for both sexes is higher than those found among our cohort when limited to individuals between 18 and 60 years (Figure 2).

### 3.3. HCMV Seroprevalence among Patients with HCMV-Associated Inflammatory-Based Disease

We further studied HCMV seroprevalence among patients suffering from an inflammatory disease in which a role of HCMV chronic infection had been suggested: CVD (aortic stenosis), SMI (schizophrenia and bipolar disease), and COVID-19. These patients and their respective sex- and age-matched controls (HD-2A, HD-2B, and HD-2C) were recruited during the 2020–2023-time frame (Figure 1). Our findings revealed that 89.83% of CVD patients, 69% of SMI individuals, and 70.37% of COVID-19 patients (regardless of hospitalization status) tested positive for HCMV infection. Additionally, when comparing COVID-19 patients based on their hospitalization status, we observed a significant difference (*p* = 0.019). HCMV seroprevalence was 1.4 times higher among hospitalized COVID-19 patients (76.92%) compared to non-hospitalized COVID-19 patients (53.33%) (see Table 3).

Besides, we did not find significant differences between patients (CVD, SMI, or COVID-19) and their respective controls cohorts (HD2-A, B, C). However, after splitting the groups by sex, a significant difference (*p* = 0.047) between males with SMI (56.25%) and male controls (HD-2B) (82.6%) was found. No differences regarding sex were found among CVD and COVID-19 patients and their respective controls (Table 3).

### 3.4. Anti-HCMV IgG Antibody Titre in the Andalusian Population

We further studied the effect of sex on anti-HCMV antibody levels in serum among HD-2 HCMV-seropositive individuals. This analysis showed that women had a generally higher titre of anti-HCMV IgG compared with men (*p* = 0.02) (Figure 3A). However, after splitting by age groups, this difference was only significant within the young individuals (*p*= 0.02) (Figure 3B). Moreover, we observed a significant increase in HCMV IgG antibody titres in male middle-aged individuals compared with young males (*p* = 0.03). Nevertheless, although not statistically significant, we observed a decrease in the HCMV IgG antibody titre in older individuals, independently of sex (Figure 3B,C).

We further extended our research by studying the levels of anti-HCMV antibodies between HCMV-seropositive patients and their corresponding control cohort (Figure 1, Table 3). No differences were found in HCMV IgG antibody titres in any of the conditions studied. Additionally, we performed a comparison of the HCMV IgG antibody titres among COVID-19 patients stratified by hospitalization status. Notably, this analysis showed a statistically significant higher HCMV IgG antibody titre in the hCOVID-19 group compared with the nhCOVID-19 patients (*p* = 0.03) (Figure 4C).

## 4. Discussion

Our study presents, for the first time in the 21st century, updated data regarding the seroprevalence of HCMV chronic infection in the Spanish population. Noticeably, the last study related to this matter dates to the end of the 20th century (1993). Although, in this work, the studied cohort only covered ages ranging from 2 to 60 years and did not include older individuals [9]. A subsequent study was performed in Spain in 1999 but was only focused on young females (2–40 years old) [10]. Furthermore, a recent meta-analysis article related to the global prevalence of HCMV chronic infection used these outdated and incomplete studies as landmark references for the Spanish population [8]. Thus, our research represents the first attempt to gather information related to HCMV seroprevalence in the Spanish adult population in the last three decades. Also, our research provides first-time HCMV seroprevalence data in Andalusia, which is a region that accounts for approximately 20% of the Spanish population. Previous reports on HCMV infection seroprevalence in Spain had focused on a smaller region, Madrid. 

Our findings indicate that HCMV seroprevalence among Spanish adults was 70.83% a decade ago and 73.48% in 2023. In contrast, the study by de Ory et al. in 1993 reported Spanish HCMV seroprevalence at 62.8%. However, it is important to note that de Ory’s cohort belonged to a central Spanish region (Madrid) and ours belonged to the southern region (Andalusia). Besides, de Ory’s cohort included children and did not encompass individuals older than 60 years [9]. Thus, if we limit de Ory’s and our cohorts to those aged 18 to 60 years, the weighted HCMV seroprevalence values would be 78.52% in 1993, 65.12% in 2013, and 68.48% in 2023. These results suggest a reduction in HCMV seroprevalence in the Spanish population from 1993 to 2013 and a stagnation in the last decade. These results align with the suggested change in Spanish HCMV seroprevalence described by de Ory et al. in 1999 [9,10]. However, there are methodological and demographic differences between de Ory’s studies and ours. The determination of anti-HCMV antibodies in 1993 and 1999 was performed using the ELISA method in contrast to the CLIA technique used in our study. Nonetheless, several studies have established that both techniques are comparable [17,18,19]. Moreover, HCMV prevalence is lower in northern Europe than in southern regions [8]. Thus, we could expect to find higher HCMV seroprevalence in Andalusia compared to Madrid. Therefore, this demographic difference might work in our favour as we could be underestimating the difference observed. Nevertheless, to rule out any bias it would be advisable to conduct further studies to assess if there are any variations in HCMV seroprevalence between the different Spanish regions. 

When comparing HCMV seroprevalence by sex, we found a higher rate in HD females compared with males, independently of the recruitment period (HD-1 and HD-2). Additionally, the HCMV IgG antibody titre analysis in the HD-2 sub-cohort revealed higher titres in females compared with male donors, but this was only statistically significant among young individuals, as described in previous Asian reports [20,21]. Moreover, HCMV seroprevalence in women of childbearing age (18–40 years) in 2023 (HD-2, 69.69%) was 1.4-fold higher than in 2013 (HD-1, 50%). Our results suggest a change in the social behaviour of young Spanish women in the 21st century, as de Ory’s studies from 1993 and 1999 suggested a decline. An increase in HCMV seroprevalence in women of childbearing age could lead to an enhanced risk of congenital defects in the fetus. 

We also studied HCMV IgG antibody titres with age, finding an increase that reached a peak in the middle-aged group, as previously reported in the British population [22], and this increase was statistically significant in males. Furthermore, our cohort included individuals older than 70 years in whom a decline was observed. High HCMV IgG antibody titres have been associated with both short- and long-term mortality rates, as well as an increased risk of hospitalization in older individuals [23]. It is worth noting that our older group predominantly consists of non-frail individuals and includes nonagenarians, which may explain the lower HCMV IgG antibody titres observed compared with the middle-aged group.

Based on the results obtained from the analysis of HCMV-related diseases, no disparities in HCMV seroprevalence or HCMV IgG antibody titres were found between the control and patient groups, as described previously [15,24,25]. In the SMI context, we found that male patients presented a significantly lower HCMV seroprevalence when compared to the male control group (HD-2B). However, the sample size of the SMI group is relatively small, making it difficult to draw strong conclusions. One explanation could be the difference in disease onset between males and females, as SMI patients are characterized by social isolation with male patients manifesting the disease at earlier ages (20–24 years) compared to females (25–30 years and ≥45 years) [26,27]. Thus, within the context of HCMV transmission, an earlier onset of the mental disorder could influence the virus’s seroprevalence, contingent upon the patient’s sex. Nevertheless, we have not found differences regarding disease initiation between male and female SMI patients in our cohort (Appendix A). Schizophrenia and bipolar disorder are intricate pathologies influenced by various factors, including the social environment, lifestyle habits, age, and exposure to recurrent pathogens such as HCMV. For instance, a recent study has shown that schizophrenia and bipolar disorder patients who experienced childhood maltreatment exhibit elevated levels of HCMV antibodies and alterations in the frequencies of immune cell subsets compared to those who did not [28]. Furthermore, congenital HCMV has been shown to play a crucial role in the development of schizophrenia only when a specific allele is present in the fetus [29]. Thus, to clarify this result, further studies in bigger cohorts are needed. Regarding COVID-19, our results show differences in HCMV seroprevalence and HCMV IgG antibody titres among patients and controls. Although not statistically significant, hCOVID-19 individuals had a higher HCMV seroprevalence (76.9%) than the nhCOVID-19 individuals (53.3%). This finding aligns with previous research, indicating a connection between HCMV chronic infection and an increased likelihood of hospitalization due to SARS-CoV-2 infection [15]. In line with this, hCOVID-19 patients showed a significant increase of HCMV IgG titres than nhCOVID-19 individuals. However, numerous studies have ruled out a correlation between anti-HCMV IgG titres and COVID-19 severity or with the risk of HCMV reactivation [30,31]. Given that HCMV IgG titres increase with age and that the mean age of nhCOVID-19 individuals was 10 years lower than that in the hCOVID-19 group, the difference observed could be attributable to age rather than to an association of HCMV and the severity of the disease.

It has been pointed out that HCMV infection might not be independently linked to all-cause mortality or cardiovascular mortality within the general population (reviewed in [32,33]. Likewise, HCMV IgG antibody titres do not seem to be linked to a higher chance of HCMV reactivation or to being more or less prone to disease in healthy individuals [31]. However, HCMV chronic infection has been shown to drive the expansion of immune cells that are associated with numerous inflammatory pathologies such as cardiovascular disease, COVID-19, and SMI [16,34,35,36,37]. In addition, chronic HCMV infection, inflammaging, and advancing age significantly affect the immune system, ultimately contributing to the onset of various diseases [11,16,38,39]. Besides, a recent study has recently shown that the response to HCMV depends on the sex of the individual [40]. Hence, it is possible that underlying factors, such as lifestyle (obesity, sedentarism, alcohol abuse, smoking, etc.), social environment, sex, and/or genetics, play a role in triggering a pathological response to HCMV infection, rather than the virus itself being the sole cause.

## 5. Conclusions

Our research suggests that in the second decade of the 21st century, the Spanish HCMV seroprevalence has not undergone changes. However, in women of reproductive age, our data suggest an upward trend shift, as we found a 1.4-fold increase in HCMV seroprevalence in the last 10 years. This shift is noteworthy due to the potential implications of congenital HCMV infection and its possible role in the development of inflammatory-based diseases. Nevertheless, our cohort is relatively small compared to similar studies, and further research is necessary to validate these findings. The precise evaluation of HCMV seroprevalence has become increasingly pivotal for public health planning and for the identification of individuals at risk. Therefore, it is crucial to consistently monitor and gather data on HCMV seroprevalence from different regions to be considered for future public health strategies and interventions.

## Figures and Tables

**Figure 1 viruses-16-00006-f001:**
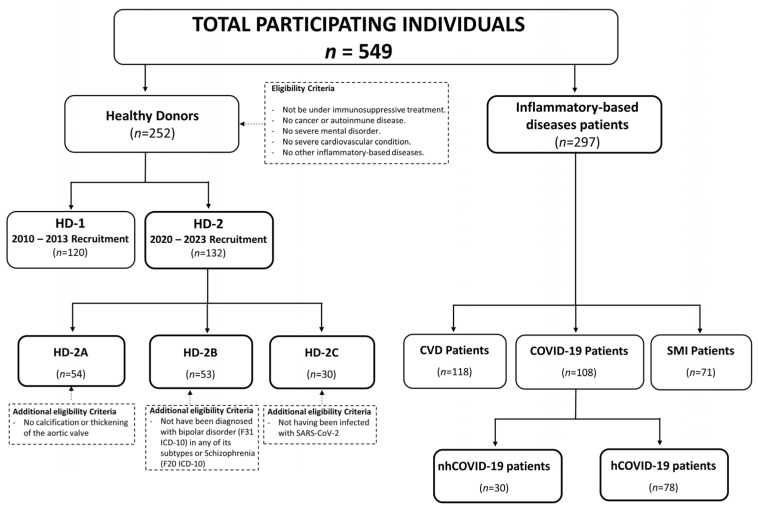
Patient recruitment flowchart, including healthy adult donors (HDs) from the 2010 to 2013 period (HD-1) and 2020 to 2023 period (HD-2), and patients with HCMV-related inflammatory diseases: cardiovascular patients (CVD), patients with serious mental illness (SMI), and COVID-19 patients. COVID-19 patients were stratified according to severity: hospitalized COVID-19 (hCOVID-19) and non-hospitalized COVID-19 patients (nhCOVID-19). HD-2 sub-cohorts were age/sex-matched with their corresponding patients’ group: HD-2A as control for CVD patients, HD-2B for SMI patients, and HD-2C for COVID-19 patients.

**Figure 2 viruses-16-00006-f002:**
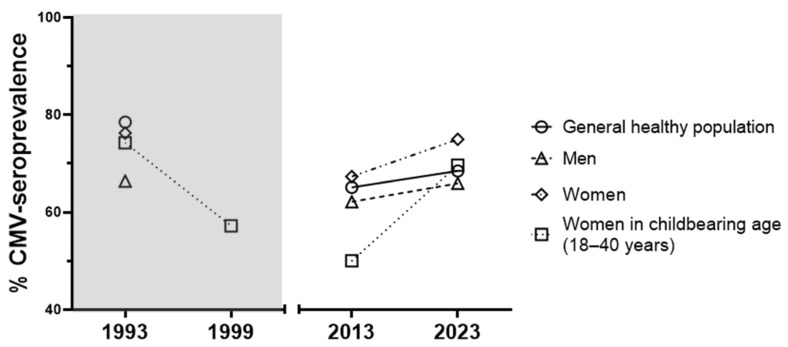
HCMV seroprevalence in adult Spanish healthy population in 1993–1999, and 2013–2023. Left panel of the graph shaded in grey shows the HCMV seroprevalence from last available (1993) general healthy population data [9], including individuals between the ages of 18 and 60 years. Right panel of the graph shows the HCMV seroprevalence from our cohort, including individuals between 18 and 60 years.

**Figure 3 viruses-16-00006-f003:**
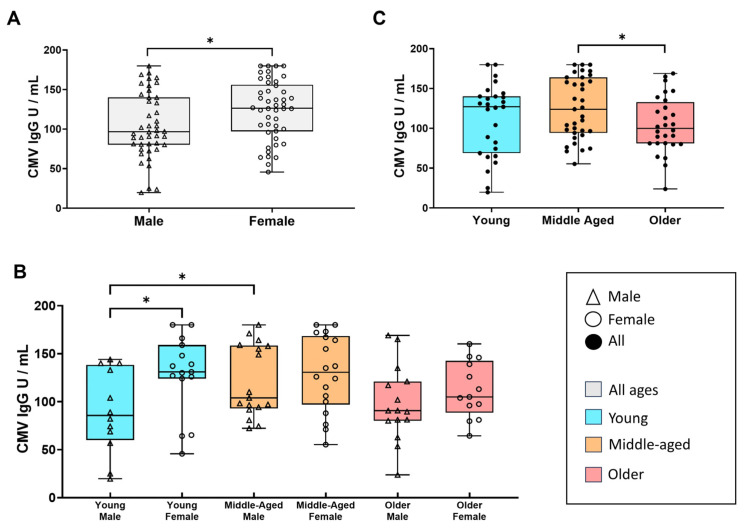
Anti-HCMV IgG antibody titre in HD-2 individuals. Boxplots represent HCMV IgG antibody units per milliliter (U/mL) in HD-2 individuals (2020–2023). (**A**) Groups segregated by sex. (**B**) Groups segregated by age and sex. (**C**) Groups segregated by age. Shapiro–Wilk test was used to determine a normal or non-normal distribution, according to this, Student’s *t*-test or Mann–Whitney U test was performed. Significant *p*-values were indicated (*).

**Figure 4 viruses-16-00006-f004:**
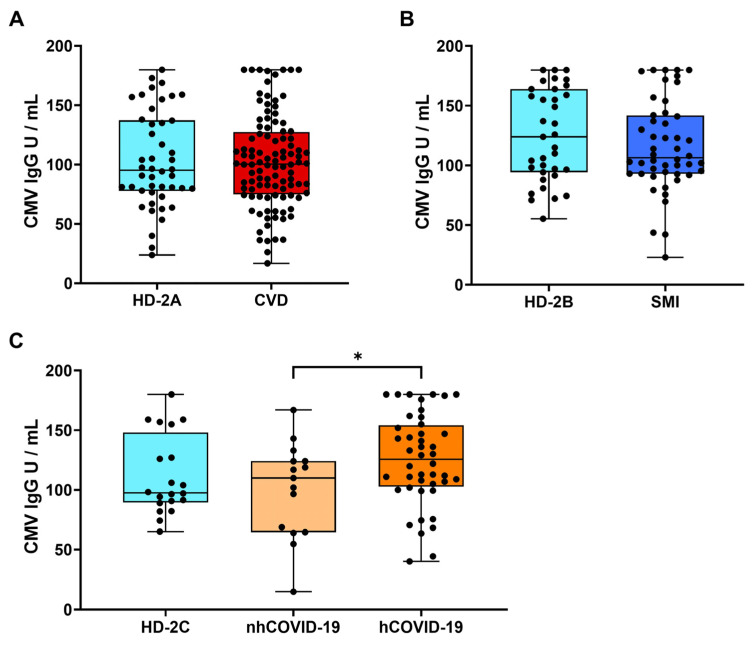
HCMV IgG antibody titre in patients. Boxplots represent HCMV IgG antibody units per milliliter (U/mL) in (**A**) CVD, (**B**) SMI, and (**C**) hCOVID-19 and nhCOVID-19 patients and their respective controls. Shapiro–Wilk test was used to determine a normal or non-normal distribution. Student’s *t*-test or Mann–Whitney U test was performed between controls and patients and hospitalized and non-hospitalized SARS-CoV-2 patients. Significant *p*-values were indicated (*).

**Table 1 viruses-16-00006-t001:** Cohorts and sub-cohorts of study participants. Informative data on the characteristics and inclusion criteria of the different cohorts and sub-cohorts of healthy individuals and patients.

Cohort	*n*	Age	Eligibility Criteria
HD-1	120	19–97 years	No cancer or autoimmune disease. No mental illness was diagnosed. No cardiovascular disease known.
HD-2	132	19–89 years	No cancer or autoimmune disease. No other conditions are known.
HD-2A	54	50–89 years	Have not been infected by the SARS-CoV-2 virus. No calcification or thickening of the aortic valve.
HD-2B	53	37–72 years	No mental illness was diagnosed.
HD-2C	30	19–88 years	No cancer or autoimmune disease. Have not been infected by SARS-CoV-2 virus. No cardiovascular disease known.
CVD	118	42–79 years	Have been diagnosed with aortic stenosis and proposed surgical valve replacement. No cancer or autoimmune disease. Have not been infected by SARS-CoV-2 virus.
SMI	71	42–68 years	Have been diagnosed with bipolar disorder (F31 ICD-10) or schizophrenia (F20 ICD-10) and required hospitalization.
Non-hospitalized COVID-19	30	22–68 years	Have been infected by SARS-CoV-2. No hospitalization is required to treat SARS-CoV-2 infection.
Hospitalized COVID-19	78	19–90 years	Have been infected by SARS-CoV-2. Required hospitalization to treat SARS-CoV-2 infection.

**Table 2 viruses-16-00006-t002:** HCMV seroprevalence in HD-1 (2010–2013) and HD-2 (2020–2023) individuals.

			*n*		Age (±SD)		HCMV Seroprevalence		*p*-Value ^2^
			HD-1	HD-2	HD-1	HD-2	HD-1	HD-2	
**Total** ^1^			120	132	47.1 (±23.1)	48.1 (±19.9)	70.83%	73.48%	0.395
		Male	48	65	41.9 (±20.5)	49.6 (±20.2)	64.58%	70.77%	0.543
		Female	72	67	50.4 (±24)	46.8 (±19.7)	75%	80.60%	0.541
	Young		49	49	26.2 (±4.09)	26.2 (±3.63)	51.02%	55.10%	0.416
	*[18–35 years]*	Male	24	24	26 (±4.56)	27.2 (±4.37)	54.17%	50%	1
		Female	25	25	25.9 (±3.90)	25.8 (±3.60)	48%	60%	0.148
	Middle Age		45	50	49.4 (±5.89)	51.8 (±6.91)	84.40%	80%	0.604
	*[36–64 years]*	Male	16	23	49.7 (±5.75)	53.5 (±7.74)	81.25%	82.61%	1
		Female	29	27	48.3 (±6.29)	50.4 (±6.21)	86.20%	77.80%	0.497
	Older		26	33	83.9 (±6.99)	75.2 (±6.03)	88.46%	90.91%	0.688
	*[>65 years]*	Male	8	18	79 (±6.04)	74.7 (±6.85)	75%	83.33%	0.330
		Female	18	15	84.4 (±7.21)	75.7 (±4.81)	94.4%	100%	1

^1^ Due to the difference in sample size, a calculation of the weighted mean age and weighted HCMV seroprevalence ratio was performed. ^2^ *p* value resulting from HCMV seroprevalence ratio comparison between HD-1 and HD-2.

**Table 3 viruses-16-00006-t003:** HCMV seroprevalence rates in HCMV-related diseases.

Disease		*n*		Age (±SD)		HCMV Seroprevalence		*p*-Value
		HD	Patients	HD	Patients	HD	Patients	
CVD		54	118	68.5 (±10.67)	66.3 (±7.84)	87.03%	89.83%	0.606
	Male	33	79	67.2 (±9.19)	65.4 (±7.95)	78.78%	87.34%	0.260
	Female	21	39	70.5 (±11.32)	68.2 (±7.25)	100.00%	95%	0.537
SMI		53	71	52.8 (±7.89)	54.4 (±7.25)	81.13%	69%	0.150
	Male	23	32	53.5 (±7.74)	53.9 (±6.12)	82.6%	56.25%	**0.047**
	Female	30	39	52.3 (±8.29)	54.7 (±8.04)	80%	79.50%	1
hCOVID-19		30	78	48.9 (±16.97)	52 (±16.08)	66.66%	76.92%	0.327
	Male	15	48	50.8 (±16.6)	51.9 (±16)	66.66%	72.92%	0.523
	Female	15	30	46.9 (±17.08)	51.9 (±16.16)	66.66%	83%	0.464
nhCOVID-19		30	30	48.9 (±16.97)	42 (±12.74)	66.66%	53.33%	0.430
	Male	15	14	50.8 (±16.6)	42.4 (±12.27)	66.66%	50%	0.462
	Female	15	16	46.9 (±17.08)	41.6(±12.75)	66.66%	56.25%	0.716
		Non Hospitalized	Hospitalized	Non Hospitalized	Hospitalized	Non Hospitalized	Hospitalized	
COVID-19		30	78	42 (±12.74)	52 (±16.08)	53.33%	76.92%	**0.019**
	Male	14	48	42.4 (±12.27)	51.9 (±16)	50%	72.92%	0.102
	Female	16	30	41.6(±12.75)	51.9 (±16.16)	56.25%	83%	0.167

## Data Availability

The data that support the findings of this study are available from the corresponding author A.P., upon reasonable request.

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
