# Peer review of "Spanish HCMV Seroprevalence in the 21st Century"

_viruses, 2023, doi:10.3390/v16010006_

Round 1

Reviewer 1 Report

Comments and Suggestions for Authors

This paper reports a study of human cytomegalovirus (HCMV) seroprevalence in Spain, using specimens collected in 2010-2013 and 2020-2023.  The results are compared with results from prior serosurveys conducted in Spain in 1993 and 1999.  In addition, the new study included comparisons of HCMV seropositivity in healthy individuals and in individuals with “CMV-associated inflammatory-based diseases”.

While some useful information has been obtained (for example, the information in Fig. 4), a number of significant problems with the analysis and presentation need to be addressed.

Major issues:

1.  The paper’s title mentions “Spanish CMV seroprevalence,” and much of the analysis involves statistical comparisons between the new results and results of prior serosurveys conducted in Spain.  The current study involves specimens collected from residents of the Andalusian region.  These data are then compared with results obtained in the Madrid region in 1993 and 1999.  Fig. 2 provides a graph comparing the two new and two older study periods.  Some potentially interesting differences are visible in the graph, but the foundation for their interpretation is shaky.  First, the older and newer studies connect to distinct regions of Spain, a point the authors suggest is significant.  This makes it impossible to make robust temporal comparisons between the two studies.  Second, serological assays for CMV assays can differ in their sensitivity and sensitivity.  Without doing some experiments to learn how the assays used in the earlier studies perform in comparison to the newer assays, there is no robust basis for detailed comparison of results obtained in the two regions.

2.  For the reasons stated above, the Fig. 2 graph should not include lines between the 1999 and 2013 results, and any inference of “trends” needs to be done cautiously, and with meaningful qualifications.  In addition, the sentences spanning lines 248-249 and 304-305 should be deleted.

Specific comments and suggestions:

1.  The journal Viruses covers viruses of humans and other forms of life.  Distinct cytomegaloviruses inhabit diverse mammalian species.  Because of this, it is better to call the virus of interest “human cytomegalovirus (HCMV)”.

2.  Line 49.  Isn’t HCMV a cause of mononucleosis?  Rather than saying that primary infection “does not cause any disease,” you might mention mononucleosis and the likelihood that primary infection likely results in non-specific symptoms in that direction.  Provide a reference.

3.  Fig. 3B - needs word labels, as used in panels A and C, in the event a reader does not have full color vision.

4.  Why not include all data points in 3C, as done in the other graphs?

5.  Start a new paragraph in line 60, after “population”.

6.  Table S1 seems suitable for inclusion in the main text, rather than as Supplementary material.

Comments on the Quality of English Language

The minor points are mentioned above.

Author Response

REVIEWER #1

This paper reports a study of human cytomegalovirus (HCMV) seroprevalence in Spain, using specimens collected in 2010-2013 and 2020-2023.  The results are compared with results from prior serosurveys conducted in Spain in 1993 and 1999. In addition, the new study included comparisons of HCMV seropositivity in healthy individuals and in individuals with “CMV-associated inflammatory-based diseases”.

While some useful information has been obtained (for example, the information in Fig. 4), a number of significant problems with the analysis and presentation need to be addressed.

Major issues:

  • The paper’s title mentions “Spanish CMV seroprevalence,” and much of the analysis involves statistical comparisons between the new results and results of prior serosurveys conducted in Spain. The current study involves specimens collected from residents of the Andalusian region. These data are then compared with results obtained in the Madrid region in 1993 and 1999. Fig. 2 provides a graph comparing the two new and two older study periods. Some potentially interesting differences are visible in the graph, but the foundation for their interpretation is shaky. First, the older and newer studies connect to distinct regions of Spain, a point the authors suggest is significant. This makes it impossible to make robust temporal comparisons between the two studies. Second, serological assays for CMV assays can differ in their sensitivity and sensitivity. Without doing some experiments to learn how the assays used in the earlier studies perform in comparison to the newer assays, there is no robust basis for a detailed comparison of results obtained in the two regions.

We thank the reviewer for their valuable comment. We agree with the reviewer's observation that making a direct comparison between our study and previous studies conducted in Spain may be inappropriate due to methodological differences.

While we acknowledge the geographical differences (Madrid vs Andalusia), we present data indicating that areas in the south exhibit a higher seroprevalence of the virus compared to those at higher latitudes [9 in text, lines 268-269]. Regarding the methodological differences, we employed the chemiluminescence technique (CLIA) which has been proven to be comparable to the ELISA technique used in the 1993 and 1999 studies in terms of sensitivity and specificity according to several publications [18,19,20 in text; lines 265-268]. Consequently, our analysis could have underestimated the statistical difference, but we still observed a decrease in the HCMV seroprevalence rate in the 21st century.

To address the reviewer's comment, we have made modifications to the results and discussion sections [Lines 254-273 (previously lines 252-253); Lines 339-340 (previous lines 303-305)], specifically addressing and discussing the differences between studies and the limitations of our analysis in that regard. We hope these changes enhance the overall clarity and robustness of our work.

  1. For the reasons stated above, the Fig. 2 graph should not include lines between the 1999 and 2013 results, and any inference of “trends” needs to be done cautiously, and with meaningful qualifications. In addition, the sentences spanning lines 248-249 and 304-305 should be deleted.

We thank the reviewer for their observation. To address this point, Figure 2 has been modified and the text regarding 248-249 and 304-305 lines has been modified.

Specific comments and suggestions:

1.The journal Viruses covers viruses of humans and other forms of life.  Distinct cytomegaloviruses inhabit diverse mammalian species.  Because of this, it is better to call the virus of interest “human cytomegalovirus (HCMV)”.

CMV terminology has been changed to HCMV throughout the manuscript.

  1. Line 49. Isn’t HCMV a cause of mononucleosis?  Rather than saying that primary infection “does not cause any disease,” you might mention mononucleosis and the likelihood that primary infection likely results in non-specific symptoms in that direction.  Provide a reference.

We express our gratitude to the reviewer for their observation. It is indeed accurate that cytomegalovirus can account for up to 5-10% of mononucleosis cases.

To address this point we have made changes in the introduction [Lines 50-52, previously line 49].

  1. Fig. 3B - needs word labels, as used in panels A and C, in the event a reader does not have full color vision.

We express our gratitude to the reviewer for raising this point. To enhance the readability of the image for all readers we have made the required modifications to the image.

  1. Why not include all data points in 3C, as done in the other graphs?

We thank the reviewer’s comment. We have now modified panel C of figure 3 to show the points corresponding to each individual.

  1. Start a new paragraph in line 60, after “population”.

This modification has been addressed.

  1. Table S1 seems suitable for inclusion in the main text, rather than as Supplementary material.

We thank the reviewer for their suggestion. We have included the supplementary Table to the main text [Line 130]. All Tables have been renamed according to this new order and their references in text adapted.

Reviewer 2 Report

Comments and Suggestions for Authors

In their manuscript “Spanish CMV seroprevalence...", Alvarez-Heredia et al. describe a relatively small cohort study (only 252 healthy donors) and 297 patients with various inflammatory diseases. The manuscript shows that CMV seroprevalence has decreased in recent years compared to the study from the early 1990s. No significant differences were found between the control groups and the COVID patients, who were found to have higher CMV antibody levels. The manuscript is well written and very easy to read, and the graphics are produced to high standards.

The main weakness of the study is the relatively small size of the control group from which the conclusions are drawn. The difference between patients with mental illness may not be as clear, but the authors make a strong point of this difference. It is interesting to note that the authors found a lower CMV seroprevalence in the 21st century than in the study that was only 20 years older (1993). The authors do not mention that the analysis (testing) was probably done differently, and do not explain how younger participants from the 1990s show lower titers in older groups 20 years later?

IN fact there are a number of studies investigating CMV seroprevalence after 1993, and several studies suggested some change in seroprevalence (e.g. Ref 10).

I.e. the conclusions are very strong and based on a limited number of participants. I think the authors should tone down their language. Discussion should include more details on SMI and similar studies. COVID-19 differences can be also additionally rationalized.

The study is not particularly innovative and simple (the patient groups are interesting), although population studies are immensely important, this manuscript is based on very small size of sampels compared to similar studies.

Minor:

Figure 1. Number of patients 428 in not correct

Figure 1. SMD instead of SMI

The title is overexaggerated and pompous for such study (in health in disease???, and the study was not decade long, samples were collected on two occasions over 10 years ).

Author Response

REVIEWER 2

Comments and Suggestions for Authors

In their manuscript “Spanish CMV seroprevalence...", Alvarez-Heredia et al. describe a relatively small cohort study (only 252 healthy donors) and 297 patients with various inflammatory diseases. The manuscript shows that CMV seroprevalence has decreased in recent years compared to the study from the early 1990s. No significant differences were found between the control groups and the COVID patients, who were found to have higher CMV antibody levels. The manuscript is well written and very easy to read, and the graphics are produced to high standards.

We thank the reviewer for their positive words.

The main weakness of the study is the relatively small size of the control group from which the conclusions are drawn. The difference between patients with mental illness may not be as clear, but the authors make a strong point of this difference.

We agree with the reviewer that the size of the cohort is a limitation which has been discussed in lines (295-312). The discussion concerning the SMI group have been rewritten and toned down pointing out the limitation of the sample size and the necessity of performing further research.

It is interesting to note that the authors found a lower CMV seroprevalence in the 21st century than in the study that was only 20 years older (1993). The authors do not mention that the analysis (testing) was probably done differently, and do not explain how younger participants from the 1990s show lower titers in older groups 20 years later?

Methodological differences between our study and those performed in the 20th century have been discussed now in lines 264-273. Lower HCMV IgG titres in the older group have been discussed in lines 286-292. Of note, de Ory’s studies do not provide any information regarding HCMV titres in their cohort. Thus, we do not completely understand what the reviewer means by “how younger participants from the 1990s show lower titers in older groups 20 years later?” since there is no information regarding this in the 90’s studies.

In fact, there are a number of studies investigating CMV seroprevalence after 1993, and several studies suggested some change in seroprevalence (e.g. Ref 10).

We thank the reviewer for their observation. We wish to emphasize there are only two studies concerning Spanish HCMV seroprevalence which were published in 2001 and 2004 by de Ory et al. Although the recruitment period in those studies occurred in 1993 and 1999, respectively (10,11). Both studies have been referenced and discussed. For more clarity, we have added a new sentence in the discussion highlighting that de Ory’s data also suggested a decrease in HCMV seroprevalence (lines 262-264).

I.e. the conclusions are very strong and based on a limited number of participants. I think the authors should tone down their language.

To accommodate the reviewer request we have now made changes to the discussion and conclusion sections to show this limitation [Lines 324-337]. Additionally, we have adjusted the tone throughout the manuscript.

Discussion should include more details on SMI and similar studies. COVID-19 differences can be also additionally rationalized.

The discussion has now been corrected to include more information regarding this point.

The study is not particularly innovative and simple (the patient groups are interesting), although population studies are immensely important, this manuscript is based on very small size of samples compared to similar studies.

We agree with the reviewer that our sample is limited and smaller than other studies. However, we would like to highlight the lack of data regarding CMV seroprevalence in Spain in the last 30 years. This absence of information makes our data relevant as it opens a crucial dialogue and underscores the necessity for more frequent studies of this nature. Such studies have the potential to furnish significant information since accumulating evidence suggests an important role of HCMV chronic infection in disease development.

Minor:

Figure 1. Number of patients 428 in not correct

We apologize for the error it has now been corrected.

Figure 1. SMD instead of SMI

We appreciate the reviewer for their observation. Regrettably, there was an error that led to the inclusion of an outdated version of the graph. We have now included the correct version.

The title is overexaggerated and pompous for such study (in health in disease???, and the study was not decade long, samples were collected on two occasions over 10 years).

We thank the reviewer for his comments on this matter. We have changed the title to 'Spanish HCMV Seroprevalence in the 21st Century'. We hope that this modification meets the reviewer's approval.

Round 2

Reviewer 2 Report

Comments and Suggestions for Authors

The revised version of the manuscript is much improved, especially the discussion and description of the results. The authors have addressed all the questions raised and provided adequate answers and explanations.

Although the study is limited with its small numbers, it is an important contribution to public health surveillance, an area of research that is immensely important but largely neglected.

As for the comment below, I meant that the same group of people who were 20 years old in the 1990s should have a similar seroprevalence 20 years later if the methodology has not changed significantly.

It is interesting to note that the authors found a lower CMV seroprevalence in the 21st century than in the study that was only 20 years older (1993). The authors do not mention that the analysis (testing) was probably done differently, and do not explain how younger participants from the 1990s show lower titers in older groups 20 years later?

 Methodological differences between our study and those performed in the 20th century have been discussed now in lines 264-273. Lower HCMV IgG titres in the older group have been discussed in lines 286-292. Of note, de Ory’s studies do not provide any information regarding HCMV titres in their cohort. Thus, we do not completely understand what the reviewer means by “how younger participants from the 1990s show lower titers in older groups 20 years later?” since there is no information regarding this in the 90’s studies.